# Shifts in controls and abundance of particulate and mineral-associated organic matter fractions among subfield yield stability zones

**Sam J. Leuthold**[1,2]**, Jocelyn M. Lavallee**[3]**, Bruno Basso**[4]**, William F. Brinton**[5]**, and M. Francesca Cotrufo**[1]

[1]Soil and Crop Sciences Department, Colorado State University, Fort Collins, CO 80523, USA
[2]Graduate Degree Program in Ecology, Colorado State University, Fort Collins, CO 80523, USA
[3]Environmental Defense Fund, New York City, NY 10010, USA
[4]Department of Earth and Environmental Sciences, Michigan State University, East Lansing, MI 48824, USA
[5]Woods End Laboratories, Augusta, ME 04330, USA

**Correspondence:** Sam J. Leuthold (sam.leuthold@colostate.edu)

**Abstract.** Spatiotemporal yield heterogeneity presents a significant challenge to agricultural sustainability efforts and can strain the economic viability of farming operations. Increasing soil organic matter (SOM) has been associated with increased crop productivity, as well as the mitigation of yield variability across time and space. Observations at the regional scale have indicated decreases in yield variability with increasing SOM. However, the mechanisms by which this variability is reduced remain poorly understood, especially at the farm scale. To better understand the relationship between SOM and yield heterogeneity, we examined its distribution between particulate organic matter (POM) and mineral-associated organic matter (MAOM) at the subfield scale within nine farms located in the central United States. We expected that the highest SOM concentrations would be found in stable, high-yielding zones and that the SOM pool in these areas would have a higher proportion of POM relative to other areas in the field. In contrast to our predictions, we found that unstable yield areas had significantly higher SOM than stable yield areas and that there was no significant difference in the relative contribution of POM to total SOM across different yield stability zones. Our results further indicate that MAOM abundance was primarily explained by interactions between crop productivity and edaphic properties such as texture, which varied amongst stability zones. However, we were unable to link POM abundance to soil properties or cropping system characteristics. Instead, we posit that POM dynamics in these systems may be controlled by differences in decomposition patterns between stable and unstable yield zones. Our results show that, at the subfield scale, increasing SOM may not directly confer increased yield stability. Instead, in fields with high spatiotemporal yield heterogeneity, SOM stocks may be determined by interactive effects of topography, weather, and soil characteristics on crop productivity and SOM decomposition. These findings suggest that POM has the potential to be a useful indicator of yield stability, with higher POM stocks in unstable zones, and highlights the need to consider these factors during soil sampling campaigns, especially when attempting to quantify farm-scale soil C stocks.

# 1 Introduction

A substantial number of cropland areas exhibit marked production variability from year to year (Basso et al., 2019; Driscoll et al., 2022). This phenomenon has been observed across scales, from global (Ray et al., 2015) to national (Renard and Tilman, 2019) down to the farm and subfield level (Maestrini and Basso, 2018). Previous work has demonstrated that interactions between climate (Leuthold et al., 2022b), geomorphology (Kravchenko et al., 2005), and edaphic characteristics (Al-Kaisi et al., 2016) are central drivers of this variability. These factors combine to create areas that are high-yielding under certain climatic conditions but low-yielding under others. The oscillating nature of these areas across time leads to their designation as "unstable," in contrast to "stable" areas, which maintain consistent yields regardless of a given growing season's weather. While some level of year-to-year yield variability is inevitable, sustained yield instability represents an important source of environmental degradation, as well as increased economic uncertainty for farmers (Basso et al., 2019).

Increasing cropland soil organic matter (SOM) stocks have been associated with decreased yield instability (Qiao et al., 2022). A regional survey of cropping systems across China found a significant negative relationship between crop yield variability and topsoil SOM concentrations (Pan et al., 2009). Other recent work has provided evidence that increasing SOM can reduce the effect that droughts have on crop production (Renwick et al., 2021), decreasing total crop losses and associated crop insurance indemnity payments by stabilizing yields over time (Kane et al., 2021). These results indicate that increasing SOM may be a viable management strategy for reducing crop yield variability. However, the bulk of these studies take place at the regional scale (e.g., Pan et al., 2009; Kane et al., 2021), and it remains unclear if this association holds true at the subfield level.

One means by which the relationship between SOM and yield stability may be better understood is to separate the bulk SOM into discrete physical fractions. Physical SOM fractions allow for increased understanding of the function and formation of SOM. In particular, particulate organic matter (POM) and mineral-associated organic matter (MAOM) have been shown to be well suited to act as indicators of SOM dynamics (Christensen, 2001; Cotrufo et al., 2019). Indeed, POM and MAOM have unique formation pathways (Cotrufo et al., 2015) and different average turnover times (von Lützow et al., 2008) and tend to provision different ecosystem services in managed systems (Lavallee et al., 2020). In agricultural lands, the balance of these two fractions often skews heavily towards MAOM, with reported global- to continental-scale averages of 71.5 %–79 % of C stored in the MAOM fraction (Lugato et al., 2021; Sokol et al., 2022), a distribution that can be attributed to a lack of consistent C inputs and frequent disturbances that can catalyze rapid POM decomposition (Lugato et al., 2021). Changes in POM

C stocks therefore have the potential to act as an important indicator of changes in ecosystem processes – several researchers have previously identified it as a sensitive predictor of agronomic function (Schipanski et al., 2010), especially at the field scale.

Given the increased sensitivity of SOM fractions to environmental and management factors (Prairie et al., 2023) and the association between SOM and yield stability, SOM fractions may be able to act as robust indicators for subfield patterns of yield variability. Further combining SOM fractions with additional factors associated with yield stability, such as topographic and soil physicochemical properties, may prove even more useful. Tajik et al. (2012) explained 96 %–98 % of the variability in soil enzyme activity using a combination of topographic and edaphic data, including SOM. Similar approaches have been applied to understand variability in soil microbial diversity and activity in row cropping systems (Kaleita et al., 2017). However, to the best of our knowledge, covariation in SOM fractions with state factors has not been explored as a means to understand spatiotemporal yield variability at the farm subfield scale.

Here, we attempt to leverage the increased insight into ecosystem biogeochemistry that physical SOM fractions provide to better understand the relationship between yields, spatiotemporal yield stability, and increasing SOM. Specifically, we asked the following question: how do POM and MAOM distribute amongst areas of different yields and yield stabilities? As increasing POM is often associated with increased nutrient processing (Daly et al., 2021), aggregate formation (Witzgall et al., 2021), and improved soil structure, we hypothesized that POM-C would be highest in high-yielding, stable areas and, consequently, that the ratio of POM : MAOM-C would be highest in these zones. Additionally, we expected these areas to have a higher amount of bulk SOM-C overall. Accordingly, we hypothesized that the lowest POM : MAOM-C values would be observed in the unstable yield zones, indicating decreased SOM biogeochemical functioning, and that these unstable areas of the field would be relatively C poor as a result of inconsistent residue inputs. To investigate these hypotheses, we performed a combined size-density fractionation on soils sampled from nine farms across the central United States. At each farm, we sampled from areas of different yields (i.e., low-, moderate-, and high-yielding) and different yield stabilities (i.e., stable and unstable). We used linear mixed-effect models to examine the distribution of SOM among physical fractions as it relates to the various stability zones and then supported our findings using a gradient boosting machine learning approach. Our goal was to understand if physical SOM fractions could be used as indicators of areas of various yield stability at the farm subfield scale and, consequently, if recommendations for improving yield stability could be gained via an understanding of fractional SOM distribution.

## 2 Materials and methods

### 2.1 Site descriptions and soil sampling

Nine farms across the upper Midwestern United States were sampled to assess the relationship between soil characteristics and crop yield stability (Fig. 1, Table 1). The 30-year mean annual precipitation (MAP) ranged from 617–702 mm yr$^{-1}$, and the average annual cumulative growing degree days (GDD) ranged from 1502–1989 (Table 1). Crop management information was obtained via communication with growers and land managers. While crop rotation varied amongst the farms, wheat (*Triticum aestivum*), soybean (*Glycine max L.*), and maize (*Zea mays*) were the dominant crops in rotation at all sites (Table 1), and all sites were sampled during either the corn or soybean phase of the rotation. Nutrient management varied by farm as well, but as these were primarily working farms, we assumed that fertilizer management represented a mixture of economic and agronomic optimum rates for the region (see Fowler et al., 2024, for further discussion). Soils were sampled as described in Fowler et al. (2024). Briefly, at each farm, three 15 cm cores from three separate replicate areas of each of the stability zones (i.e., low-stability, medium-stability, high-stability, and unstable) were taken, such that 12 samples were collected from each farm ($n = 108$). Samples were then sent to Woods End Laboratory for characterization and analysis (Fowler et al., 2024).

### 2.2 Delineation of yield stability zones

Yield data were obtained using real-time kinematic global positioning system-enabled (RTK-GPS-enabled) research combine with a yield monitor, which recorded crop yield at a 1 m resolution. Following Maestrini and Basso (2018), raster yield data were scaled to the annual field average, and a yield level was calculated relative to the mean yield (i.e., high, medium, and low yields). Yield stability was then derived from the standard deviation (SD) of yield levels over time, with a SD value greater than $\pm 15\%$ designated as unstable. A full description of the delineation of yield stability zones is provided in Fowler et al. (2024). In the work presented here, in order to arrive at a unitless yield stability metric that accommodated differences in crop rotation, we also normalized the annual yield from the different zones by the yield in the high and stable zone for each year. We then used these normalized values to calculate zonal means and SD for the entire experimental period at each stability zone within each farm and then calculated a coefficient of variation for yields by dividing the SD by the mean. The resulting metric (yield CV) was well aligned with the stability zones derived from the SD as in Fowler et al. (2024), which we interpreted as a validation of this approach of quantifying stability numerically.

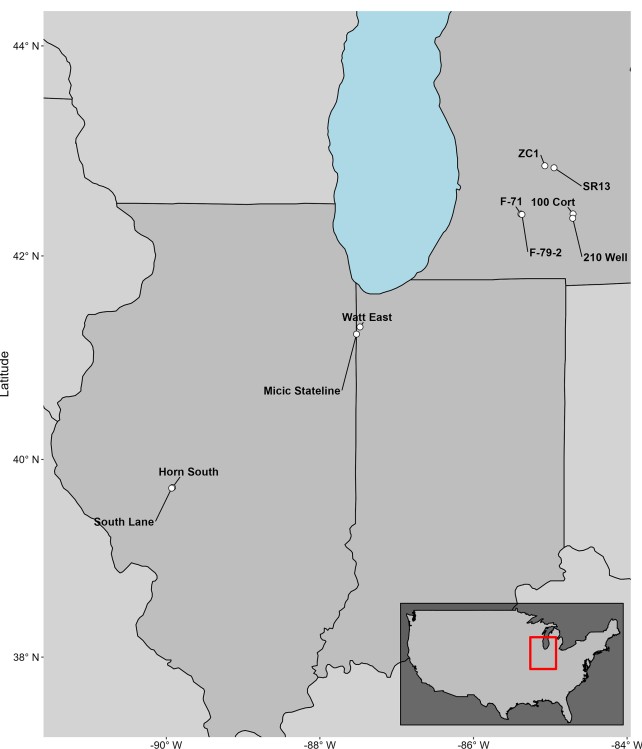

**Figure 1.** Location of farms sampled for this analysis. Sampling took place in the United States, namely Indiana, Illinois, and Michigan. Inset map shows the geographic location of the sampling region.

### 2.3 Soil processing and characterization

At Woods End Laboratory, the air-dried samples were sieved to 2 mm to remove root fragments and rocks and were ground prior to analyses. Soils were analyzed for a range of properties, including soil pH using a 1 : 1 soil : water extract and pH electrode method, Mehlich I- and Mehlich III-extracted nutrients (NCERA-13, 2015), and cation exchange capacity (NCERA-13, 2015). Total soil organic C was measured via dry combustion at 900 °C using a Shimadzu TOC-L coupled to a solid-sample-dry-combustion module (SSM-5000A) (Shimadzu Corporation, Kyoto, Japan), following manufacturer protocols (Shimadzu, 2017). Across sites, soil pH varied by almost an order of magnitude, ranging from 5.68–6.51 (Table 1). Total soil organic C (SOC) varied amongst sites as well, ranging from 8.62–25.9 g C kg soil$^{-1}$ (Table 1). Given our interest in the role of the fine fraction of soil particles in determining SOC dynamics, we did not measure texture directly. Instead, we inferred the proportion of silt and clay particles and the proportion of sand particles from the results of our physical fractionation analysis (see below). In the soil we analyzed, the distribution of particle sizes ranged from relatively sandy soils ($\sim 64\%$ sand) to soils dominated by fine particles ($> 90\%$ silt and clay particles; Table 1).

**Table 1.** Major characteristics of farms from which soils were sampled for analysis. Growing season precipitation (GSP) and mean cumulative growing degree day (cum. GDD) values represent 30-year means retrieved from the Gridmet data product.

| Farm TS1 – | GSP (mm) | Mean cum. GDD – | Soil organic carbon (g C per kg soil) | Soil pH – | Sand (%) | Silt + clay (%) | Cropping system history[*] – |
|---|---|---|---|---|---|---|---|
| 210 Well | 617 | 1522 | 11.0 | 6.49 | 64.2 | 35.8 | C/S/C/S/W |
| F-71 | 701 | 1561 | 8.62 | 5.86 | 64.3 | 35.7 | S/C/S/W |
| F-79-2 | 698 | 1559 | 9.66 | 5.68 | 64.5 | 35.5 | W/S/C/S/W |
| Horn South | 637 | 1988 | 25.9 | 6.22 | 5.2 | 94.8 | C/S/C/S/C |
| Micic State | 702 | 1753 | 16.5 | 6.00 | 52.2 | 74.8 | S/C/S/C/S |
| South Lane | 643 | 1989 | 21.1 | 6.25 | 7.4 | 92.6 | C/C/S/S/C |
| SR13 | 624 | 1513 | 13.6 | 6.21 | 39.5 | 60.5 | W/S/C/S/W |
| Watt East | 687 | 1741 | 20.1 | 6.51 | 25.2 | 74.8 | C/S/C/C/S/C |
| ZC1 | 644 | 1502 | 15.7 | 5.76 | 45.0 | 55.0 | C/S/C/S/W |

[*] C – corn (*Zea mays*); soybean (*Glycine max L.*); wheat (*Triticum aestivum*). Reported for years available during study period which ended in 2020.

## 2.4 Physical fractionation and SOM analysis

A well-homogenized subsample from all 108 samples was shipped to Colorado State University, where they were fractionated into physical SOM fractions using a combined size density fractionation as reviewed in Leuthold et al. (2022a). Briefly, a 6 g subsample of 2 mm sieved soils was dried overnight and then shaken in deionized (DI) water for 15 min. The sample was then centrifuged and the supernatant was subjected to vacuum filtration using a 20 µm nylon filter to isolate the dissolved organic matter fraction (DOM). Following the removal of the DOM, sodium polytungstate (SPT; $Na_6[H_2W_{12}O_{40}]$) adjusted to a density of $1.85\,\mathrm{g\,cm^{-3}}$ was added to each sample and samples were shaken for 18 h. After shaking, the samples were centrifuged and the supernatant was aspirated via vacuum filtration using a 20 µm nylon filter. The material that remained in the filter was characterized as the POM fraction. The remaining mineral fraction was resuspended in DI water and centrifuged and the resulting supernatant discarded. This rinsing procedure was repeated two more times to remove any residual SPT from the samples. Upon the fourth resuspension of the mineral fraction, the samples were separated via wet sieving into the coarse, heavy associated organic matter fraction (CHAOM; $> 53\,\mu m$) and MAOM fraction ($< 53\,\mu m$). These samples were then placed in an oven at 60 °C and dried to a constant weight.

Oven-dried samples were weighed, and mass recovery was assessed. If the recovered mass was not between 95 % and 105 % of the initial sample mass, the fractionation was repeated until an appropriate mass was reached. Two samples did not achieve acceptable recoveries despite repeated efforts; we do not include these samples in the analysis presented here. After weighing, samples were ground into a fine powder using a mortar and pestle and analyzed for C and nitrogen (N) concentrations via a VELP CN802 Carbon/Nitrogen Analyzer (VELP Scientific, Deer Park, NY). As the soils contained negligible content of inorganic C, total C values obtained through elemental analysis reflect fraction organic C. As the DOM fraction typically represents a minor fraction of the total SOM, especially in agricultural soils, we opted not to analyze it further after separation. However, we did find it necessary to separate the DOM and CHAOM fractions to isolate the POM and MAOM pools that are most homogenous in the composition and are best aligned with their conceptual definitions.

## 2.5 Topography and climate data acquisition

To create the data set used in subsequent multivariate analyses, several databases were called upon using R version 4.2.2 (R Core Team, 2022). For each farm, 1 km gridded, daily climate data were retrieved for the period between 2015 and 2020 from the Daymet daily surface weather data set (Thornton et al., 2022) using the R package daymetr (Hufkens et al., 2018). Climate data were trimmed to represent the average period of planting through harvest for major row crops in the northern Corn Belt, starting on Julian day 121 and ending on Julian day 304. The Daymet data set provides daily maximum and minimum temperatures, which we averaged to construct a daily mean temperature. We calculated growing degree days with a base temperature of 10 °C, such that heat units only accumulated when mean air temperatures exceeded 10 °C. Elevation data were retrieved from the USGS 3D Elevation Program using the get_tiles function contained within the terra package in R (Hijmans, 2022). To better capture landscape-level trends, we resampled and aggregated the elevation data using a factor of 8 based on an iterative analysis of the scale at which geomorphic features in agricultural land were most pronounced. We then used the terrain function in the terra package to calculate slope and topographic position indices (TPI), a measurement of the elevation of a given grid cell relative to the grid cells surrounding it (higher values indicate a relatively higher grid cell and vice versa).

## 2.6 Data analysis

### 2.6.1 Mixed linear model and regression analysis

We used mixed-effect linear models to test for differences in soil texture, soil pH, SOM-C, POM-C, and MAOM-C among stability zones. Models were built using the lme4 package in R (v.1.1-31; Bates et al., 2015) and evaluated using the anova() function in the stats package (v. 4.2.2; R Core Team, 2022). Pairwise comparisons were made using examinations of the estimated marginal means of each stability zone contrasts pair using the emmeans package (Lenth, 2022). In the models we examined, stability zone was a fixed effect and farm was a random effect. The alpha level for significant differences was set to 0.05. We also examined relationships between soil texture and MAOM-C and POM-C, as well as average crop yield and yield CV and soil texture and topographic indices via linear regression, using the lm() function in the stats R package (R Core Team, 2022). To account for differences in cropping rotation, edaphoclimatic characteristics, and management history, we scaled the response variables to the farm level using $z$ scores, with the mean and standard deviation of the population calculated at the site level, similar to the approach of Aksoy et al. (2016). All results presented below have been scaled using this method unless explicitly noted.

### 2.6.2 Assessing feature importance via gradient boosting

We used a supervised classification machine learning approach (i.e., regularized gradient boosting) to identify and understand potential indicator variables for delineating yield stability zones. We reclassified the stability zones into binary categories (i.e., stable and unstable) and then randomly split the data set into training and testing data sets such that 80 % of the data were contained in the training data set and 20 % in the testing data set. We included all SOM fraction data in the model matrix, as well as soil physical property data, nutrient data, and topography data. After training the model, we tested the optimized model on the training set and assessed its accuracy in classifying yield zones based on soil and topography characteristics. We repeated this process 1000 times to arrive at an average model accuracy. From this ensemble of gradient boosting results, we extracted the edaphic, biogeochemical, and topographic variables that had the largest and most consistent impact on zone delineation.

## 3 Results and discussion

To better understand the relationship between increasing SOM and spatiotemporal yield heterogeneity, we fractionated soils from areas of different yields and yield stabilities across nine farms in the upper Midwestern US. Our experimental goals and hypotheses were informed by a growing body of literature that shows that increasing SOM correlates with increased yield stability (Kane et al., 2021; Pan et al., 2009; Williams et al., 2016). Given that different physical fractions of SOM are thought to have different functions, we predicted that examining the fractions would give increased insight into the mechanisms behind the SOM-yield stability relationship. Specifically, we hypothesized that total SOM would be highest in high-yielding, stable areas and that these areas would have a higher POM : MAOM-C value than areas of the field that were more unstable. Contrary to our hypotheses, however, we found that, on average, unstable zones had higher SOM-C stocks than stable yielding zones (Fowler et al., 2024). Additionally, while we found significant differences in the amount of POM- and MAOM-C among different stability zones, we did not observe significant differences in POM : MAOM-C. These findings challenge the often-assumed direction of causality whereby higher SOM supports higher-yielding (Bauer and Black, 1994; Oldfield et al., 2022; Ma et al., 2023) and more stable cropping systems (Williams et al., 2016). Instead, our results indicate that causal linkages between SOM and yield stability may be bi-directional depending on the scale of the inquiry (e.g., field vs. county vs. region). Here, we present the results of our study, discuss potential drivers of POM and MAOM storage at the subfield scale, and evaluate the implications of our findings in regard to soil health indicators and soil C storage goals.

### 3.1 Patterns of C storage in bulk SOM and fractions

When examined across the farms in our study, we observed significant differences in the total SOM-C, MAOM-C, and POM-C storage between our four different yield stability zones. Post hoc analysis of the differences between zones in SOM-C indicated that unstable zones had 25.7 % more SOM-C on average than low-yielding stable zones ($p = 0.016$; Fig. 2a). We also observed trends towards increased SOM-C in the unstable zones relative to the other two stable zones based on estimated marginal means; however, these differences were not significant ($p = 0.475$ and $0.331$, respectively; Fig. 2a). These observations run in contrast to current paradigms that indicate increasing crop yield stability in response to increasing SOM content. For instance, Pan et al. (2009) showed that the average yield variability decreased by 5 % for every 1 % increase in SOM (i.e., SOM-C $\times$ 1.72) when examined at the regional scale across Chinese provinces. Although our data do not yield robust regressions between SOM-C and yield CV, the general trend of the data indicates increasing variability as standardized SOM increases (Fig. S1 in the Supplement; $p = 0.077$, $r^2 = 0.02$). The MAOM-C results mirror the patterns we observed in SOM-C, with significant differences among stability zones ($p = 0.023$; Fig. 2b) and post hoc tests indicating that the low-yielding stable zone had significantly less MAOM-C than the unstable zone ($p = 0.013$) while the medium- and high-yielding zones were not significantly different than un-

stable zones ($p = 0.522$, 0.662; Fig. 2b). The consistency of these two patterns is not surprising, as MAOM-C accounted for ~77 % of the SOM-C in our data on average, which is consistent with previous examinations of agricultural soils (Sokol et al., 2022). Subfield variation in MAOM-C content has been observed in previous studies (Usowicz and Lipiec, 2017), but to the best of our knowledge, the covariation with yield stability such as this has not been examined thus far.

The POM-C content showed a different pattern across yields and stability zones from the patterns we observed for SOM-C and MAOM-C contents (Fig. 2). While we still found a significant difference between stable and unstable zones, stable zones showed similar POM-C levels regardless of average yield ($p = 0.019$; Fig. 2c). This contrasts with our original hypothesis that POM-C would be highest in high-yielding, stable zones as a result of increased organic inputs in the form of both aboveground crop residue and crop roots (Gosling et al., 2013; Sokol et al., 2019). Our results indicate that POM-C abundance was not driven by yield or residue inputs. Rather, other characteristics of unstable yield zones may have led to increased POM-C stocks relative to stable areas.

## 3.2 Evaluation of POM : MAOM-C ratio as an indicator of biogeochemical function

The POM : MAOM-C ratio has been suggested as a useful indicator for soil biogeochemical function (Cotrufo et al., 2019). Previous work has shown that the balance of POM-C to MAOM-C may help to both explain the mineralization rate of organic N (Daly et al., 2021; Grandy et al., 2022) and identify soils that are approaching mineral C saturation sensu Stewart et al. (2007) (Castellano et al., 2015; Just et al., 2023). It has also been posited that a decreasing POM : MAOM-C ratio may indicate when residue inputs outpace residue decomposition, assuming fragmentation processes are not inhibited (Just et al., 2023). Despite similar POM-C levels across the various stable yield zones and covariation in yield with MAOM-C in our data, however, we observed no significant differences in the POM : MAOM-C ratio across the different yield stability zones ($p = 0.170$; Fig. S2). This similarity across zones, despite differences in POM- and MAOM-C among different stability zones, is likely the result of the reasonable expected effect size of changes in POM : MAOM-C ratio in agricultural soils that are inherently MAOM-C rich and POM-C poor (Prairie et al., 2023; Lugato et al., 2021). TS2 To exemplify this issue, across all of our samples, the average POM-C was 1.9 g C per kg soil, and the average total SOM-C was 14.8 g C per kg soil, with an average POM : MAOM-C ratio of 0.80. If POM-C was increased by 50 %, to 2.8 g C per kg soil, SOM-C would increase by 6.4 %, but the POM : MAOM-C ratio would adjust only to 0.78, which is an ecologically insignificant shift. As such, we were unable to use it as a metric to understand differences in SOM formation and function among different

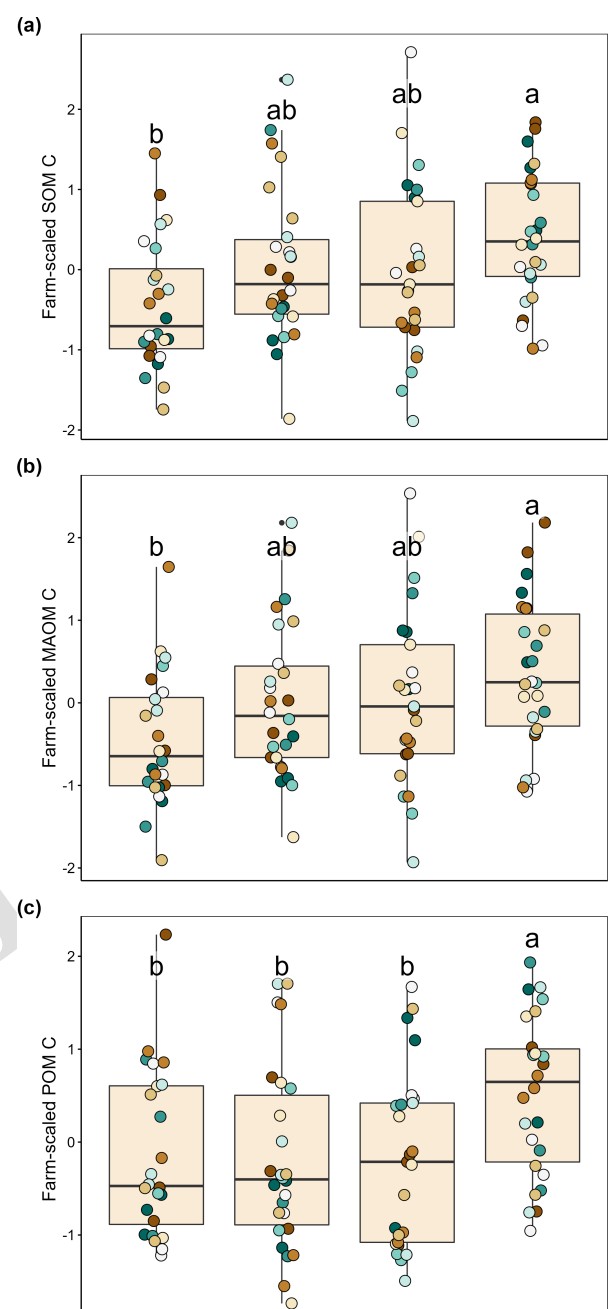

**Figure 2.** Normalized concentrations of total soil organic carbon (SOM-C) **(a)**, mineral-associated organic matter carbon (MAOM-C) **(b)**, and particulate organic matter carbon (POM-C) **(c)** among the various stability zones. Different colored points represent different farms. To account for edaphoclimatic differences among farms, we scaled all data using $z$ scores prior to analysis, with the mean and standard deviation calculated at the farm level, yielding a unitless metric to compare with. Different lower-case letters indicate significant differences ($p < 0.05$). Points are offset horizontally to improve readability of the plot.

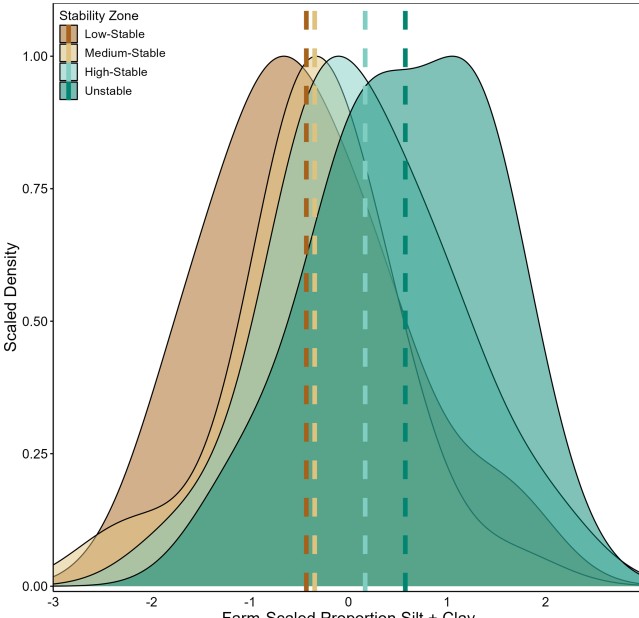

**Figure 3.** Scaled density plot of $z$-scored silt and clay content of soils from various stability zones. Dashed lines indicate the mean $z$ score for the corresponding stability zone.

stability zones. We suggest that for POM : MAOM-C ratio to be a useful indicator, further research is needed regarding how to best contextualize and understand this measure when applying it to agricultural systems. For example, exploring the ratio of POM-C to exchangeable MAOM-C may present a more actionable indicator if robust methods of exchangeable MAOM-C can be identified (Daly et al., 2021; Jilling et al., 2018)

### 3.3 Drivers of carbon accumulation in MAOM

To explore the possible mechanisms behind the variability in MAOM-C content among different yield stability zones, we examined a set of edaphic characteristics known to be well correlated with MAOM storage (Hassink, 1992). As expected, MAOM-C had a significant positive relationship with the proportion of fine particles within a sample ($p < 0.001$, $r^2 = 0.30$; Fig. S3). This relationship partially explained the patterns we observed in MAOM-C across stability zones (Fig. 2b); when we examined the difference between MAOM-C in unstable and low-yielding, stable zones as a function of the difference in fine particles, the differences in MAOM-C became negligible when particle distributions were similar ($p = 0.002$, $r^2 = 0.297$; Fig. S4). On average across sites, however, we observed significant differences in the proportion of silt and clay particles across stability zones, with unstable and high-yielding, stable zones having significantly higher farm-scaled silt and clay values than medium- and low-yielding stable zones ($p < 0.001$; Fig. 3). We posit that these differences are a result of the topographic

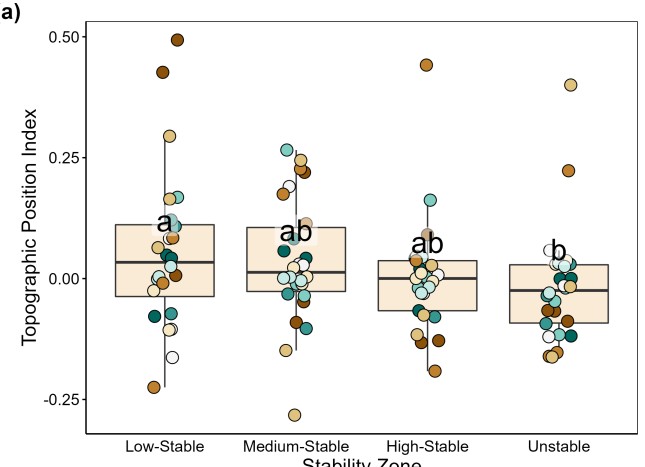

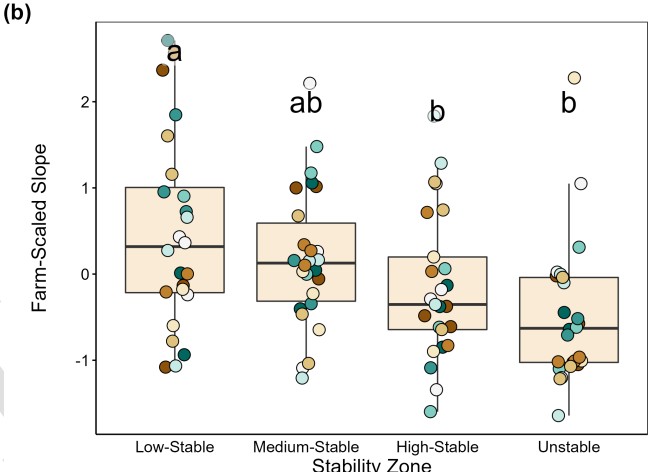

**Figure 4. (a)** Topographic position index values across the various stability zones. Lower values indicate a point in space lower than its surroundings, while higher values indicate relatively elevated locations. **(b)** Slope values scaled using $z$ scores to the individual farm level. Different colored dots represent different farms. Different lower-case letters indicate significant differences ($p < 0.05$). Points are offset horizontally to improve readability of the plot.

settings of the unstable and high-yielding, stable zones, corroborated by our observation of a significant relationship between the silt and clay content in soil and the TPI ($p < 0.001$, $r^2 = 0.10$). As observed by Maestrini and Basso (2018), both high-yielding, stable zones and unstable zones are often located in depositional areas that receive downslope contributions of fine soil particles on the decadal timescale. Our study finds a similar distribution of stability zones amongst low-lying areas, with both the high-yielding, stable zones and the unstable zones having a relatively high fine-particle content (Fig. 3), and primarily in lower areas of the field (Fig. 4). Our findings thus corroborate previous observations of crop yield heterogeneity in areas of topographic complexity (Kravchenko and Bullock, 2000; Maestrini and Basso, 2018; Leuthold et al., 2021). The similarity in texture and to-

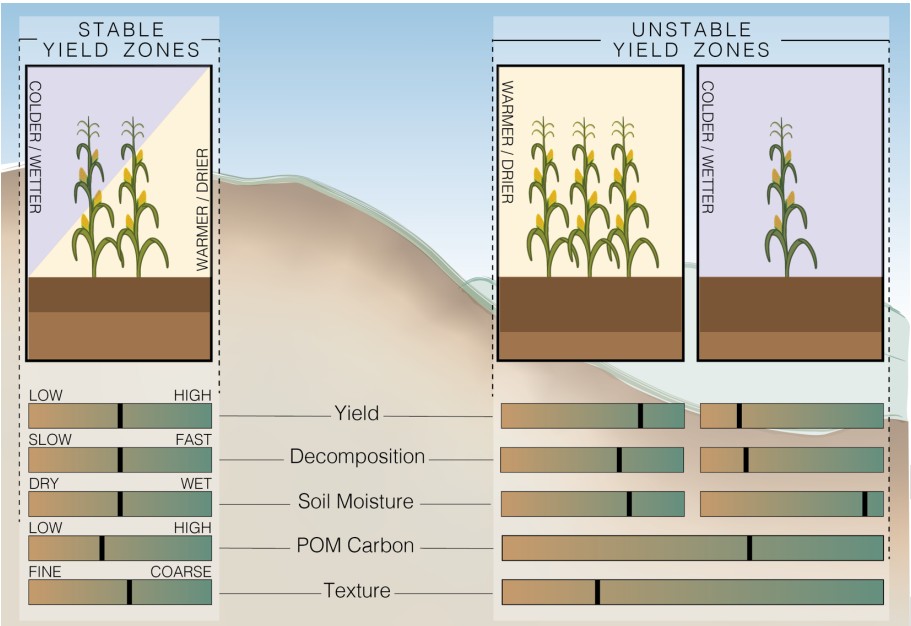

**Figure 5.** Conceptual model of proposed biogeochemical differences between stable and unstable yield zones. Stable zones, often in upslope positions, are characterized by consistent yields across weather conditions, partially due to a decreased incidence of saturated conditions. The same conditions allow for a consistent rate of particulate organic matter (POM) decomposition and nutrient provisioning across growing seasons. In contrast, low-lying, unstable yield zones experience much more drastic shifts in their soil water status from year to year, altering the rate of POM decomposition. In wetter years in which saturated soil conditions are more likely, decomposition rates are slowed, leading to an accumulation of POM. Under dry or optimal conditions, low-lying, deeper, more finely textured soils that are characteristic of the unstable yield areas may maintain increased water retention, leading to an advanced rate of nutrient cycling; increased productivity relative to the rest of the field; and, subsequently, increased inputs to the POM pool. These different mechanisms between dry and wet conditions, working in tandem, may thus create a feedback that leads to the accumulation of POM in unstable areas that we observe.

pography between unstable and high-yielding, stable zones highlights an important point: unstable zones have a capacity to be especially high-yielding under the right environmental conditions (Fowler et al., 2024; Martinez-Feria and Basso, 2020). Further research into the properties of these unstable zones, such as impeded soil drainage or impediments to rooting ability, may help to disentangle what makes some areas with similar textures and topography consistently high-yielding, while others are especially sensitive to environmental stressors.

In addition to these edaphic controls, we also observed an association between increasing yields and increasing MAOM-C content. Linear regression analysis indicated that as mean yield within a stability zone increased, so did MAOM-C. This relationship was weak, likely reflecting the influence of cropping system, soil physicochemical properties, and climate on variability in MAOM-C and yield ($p = 0.048$, $r^2 = 0.08$; Fig. S5). However, this result does mirror recent studies that support causal linkages between increasing productivity and increased MAOM-C (Prairie et al., 2023; King et al., 2023; Hansen et al., 2024). Indeed, there are a number of pathways by which increasing yield (and the associated increase in crop residue) can increase the amount of MAOM-C, especially in areas of the field that are receiving inputs of eroded minerals that may have a high capacity for sorption of dissolved organics (Van Oost and Six, 2023). Understanding relationships between topography, productivity, and mineralogy is thus key for understanding how MAOM-C accumulates in croplands and how sampling strategies must be designed to capture accurate estimates of cropland SOC stocks and their spatial variability.

### 3.4 Mechanisms for increased POM-C storage

In contrast to MAOM-C, POM-C did not closely follow the patterns of total SOM-C content variation observed across our yield stability zones, and POM-C content in unstable zones was significantly higher than in all stable zones ($p = 0.019$) that had the same POM-C content independent of yields (Fig. 2c). Further diverging from patterns in MAOM-C, we could not identify any edaphic or cropping system properties that helped to explain patterns in POM-C – it was not correlated with average yield ($p = 0.337$, data not shown), even when accounting for differences in cropping system (reported in Table 1). We did observe significant positive relationships between POM-C and soil texture and POM-C and soil pH ($p < 0.001$ and $p = 0.009$, respectively; data not shown), both of which have been suggested as poten-

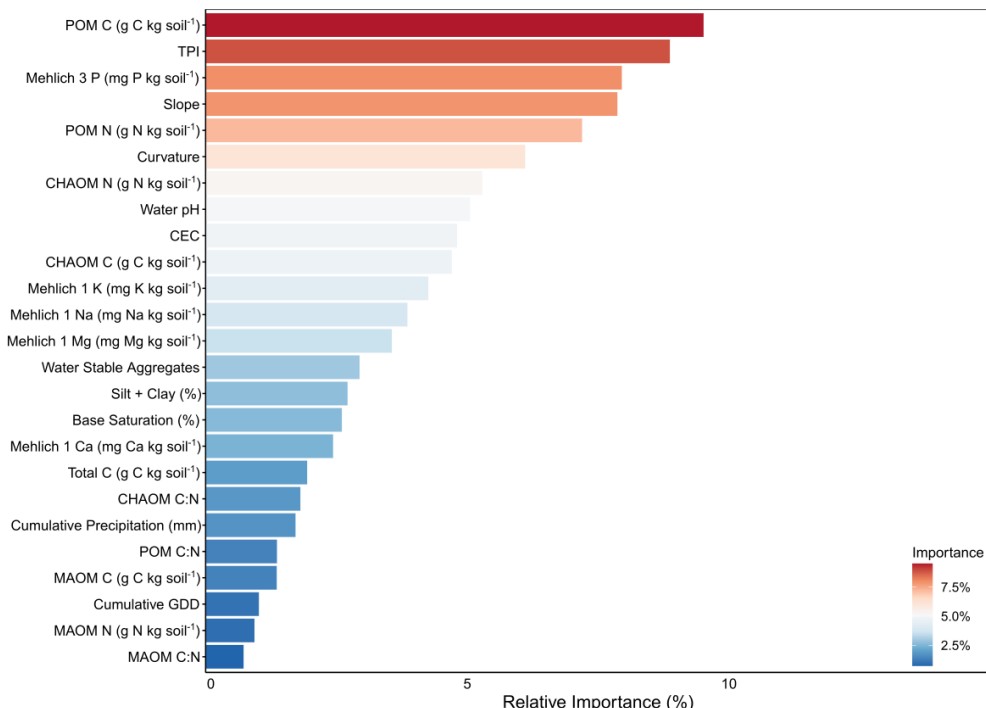

**Figure 6.** Relative importance of variables in predicting yield stability zones as determined by the gradient-boosted random forest model employed here. Relative importance represents the average of the feature importance over the course of 1000 model iterations (POM – particulate organic matter; CHAOM – coarse, heavy organic matter; MAOM – mineral-associated organic matter; C – carbon; TPI – topographic position index; P – phosphorus; N – nitrogen; CEC – cation exchange capacity; K – potassium; Na – sodium; Mg – magnesium; GDD – growing degree days).

tial controls on POM-C accumulation (Hansen et al., 2024; Kögel-Knabner and Amelung, 2021). In our study, however, our observation of the parallel increasing of POM-C and texture may be indicative of covariation in these measurements across our stability zones rather than a signal of a causality. Additionally, pH did not vary systematically across the stability zones in a way that would help to explain the patterns of POM-C variation ($p = 0.224$).

Given that we did not observe evidence that POM-C conferred additional stability to cropping systems, as posited in our original hypothesis, we propose an alternative hypothesis consistent with our findings (Fig. 5): the accumulation of POM-C within areas of increased spatiotemporal yield heterogeneity is controlled by constraints on decomposition outpacing those on productivity during the unfavorable climate years. Unstable zones tend to have a higher incidence of water-logging (Leuthold et al., 2021; Maestrini and Basso, 2018), which can impact both soil microbial activity and crop productivity. While these hydrologic stressors are most often discussed in regard to their detrimental effect on crop yield (i.e., early season saturation can reduce crop root viability and emergence) (Wenkert et al., 1981), limited oxygen availability under saturated conditions also reduces the capacity of the microbial community to depolymerize and break down POM and alters microbial community structure (Bowles et

al., 2018; Cates et al., 2022). This reduced decomposition could contribute to multi-season feedback in these unstable zones; under optimal conditions, decomposition and nutrient mineralization may be increased relative to stable yield areas due to the increased amount of available POM-C. The ecosystem services provided by POM (e.g., increased aggregation and nutrient availability), combined with optimal soil–water conditions, could then encourage increased productivity and yields, potentially leading to replenishing POM stocks that may have been decreased over the course of the optimal season (Fig. 5). This hypothesis is partially corroborated by the topographic characteristics of our various stability zones (Fig. 4). We examined both the farm-scaled TPI and the average slope for each stability zone and found that high-yielding, stable zones and unstable zones both had significantly lower average slopes and significantly lower TPI values than low- and medium-yielding stable areas ($p = 0.001$ and $p < 0.001$, respectively; Fig. 4). Additionally, we observed a significant negative relationship between the position index and POM-C; as TPI increased, the amount of POM-C decreased ($p = 0.004$; Fig. S6). We did not observe the same relationship with slope ($p = 0.903$), which could be due to different topographic positions having similar slopes (e.g., summit and footslope positions). Some evidence supporting this possibility is offered by the results of including

an interaction term between slope and stability zone, which yields a significant result ($p = 0.029$). Fowler et al. (2024) found similar relationships; they indicated that topographic variables such as slope and the log of flow accumulation (i.e., topographic wetness index) were adequate predictors of SOM-C and other soil health indicators in these fields, potentially highlighting how POM-C can act as an indicator of soil biogeochemical dynamics. These important trends lend some credence to our framework (Fig. 5), though a considerable amount of variance in POM-C remains unexplained by these factors ($r^2 = 0.07$ for TPI and $0.09$ for slope $\times$ stability zone, respectively).

Our gradient boosting analysis also supports our alternative hypothesis for increased POM storage in unstable yield zones. We used a gradient-boosted random forest analysis to determine important predictors of stable vs. unstable yield zones. Our model was able to predict stable vs. unstable zones with $\sim 72\%$ accuracy and over the course of 1000 iterations identified POM-C and TPI as the most important variables in characterizing the zones (Fig. 6). Thus, while our original hypothesis of POM acting to increase yield stability may not be supported by these data, our results do show that the POM-C content may serve as a useful and important indicator for areas prone to increased heterogeneity or variability in decomposition status when robust subfield sampling is employed. Future work that links soil microclimate data to decomposition rate across areas of complex topography will help to further elucidate the drivers of fractional SOM distribution among yield stability zones. Microbial community composition may also help provide insight into the means by which different fractions of organic matter accumulate and persist in heterogenous areas.

## 4   Conclusions

We examined the difference in C storage in bulk SOM and among SOM physical fractions in soils sampled from different yield stability zones from nine farms across the upper central United States. Whereas previous analyses have found increased crop yield stability with increasing SOM, our results indicate that this relationship may not always hold true depending on the scale of inquiry. At the subfield scale, we found increased POM-C in unstable yield zones, which may be an indicator of reduced decomposition. We also found increased MAOM-C in unstable zones, which was well correlated with increased incidence of fine particles and increased yield potential in these zones. Given the continued development of precision agriculture technologies (Basso and Antle, 2020), increased understanding of the mechanisms that confer stability unto a given area within a crop field is of paramount importance. Our work does not stand in contrast to previous publications that show that cropping system heterogeneity may be reduced with increasing SOM concentrations but instead offers an important insight that the controls on the relationship between SOM and stability depend on the scale of inquiry and that, at the subfield scale, unstable zones may be characterized by increased SOM-C, especially in the POM fraction. Our results provide steps towards understanding how geomorphology, inter-annual weather variability, and cropping system productivity interact to determine the distribution of SOM among physical fractions across heterogenous crop fields and can serve to improve nutrient management strategies and carbon sequestration objectives and guide robust sampling for the quantification of farm-scale SOM stocks.

**Code and data availability.** The data and analyses that support these findings will be made available in response to a reasonable request but are not hosted in an online repository at this time in order to protect the privacy of growers.

**Supplement.** The supplement related to this article is available online at: https://doi.org/10.5194/soil-10-1-2024-supplement.

**Author contributions.** Conceptualization: SJL, BB, MFC, JML. Data curation: SJL, BB, WFB. Formal analysis: BB, SJL. Funding acquisition: SJL, JML, MFC. Investigation: SJL, BB, WFB. Supervision: JML, MFC. Visualization: SJL. Writing (original draft preparation): SJL. Writing (review and editing): BB, WFB, JML, MFC. All authors have read and agreed to the published version of the manuscript.

**Competing interests.** M. Francesca Cotrufo is the cofounder of Cquester Analytics LLC, a service analytical facility which provides the analyses of SOM, POM, and MAOM. Bruno Basso is the cofounder and Chief Scientist of CiBO Technologies, a precision agriculture technology and data analytics firm. William F. Brinton is the founder of Woods End Laboratory, a service analytical facility that provides analyses of soil physicochemical properties. At least one of the (co-)authors is a member of the editorial board of *SOIL*. The peer-review process was guided by an independent editor.

**Acknowledgements.** The authors would like to thank Sam McNeil, Paige Lewis, Michelle Haddix, and Rebecca Even for their assistance in the lab, as well as the staff of Woods End Laboratory for their analytical assistance. We are also grateful to Erica Patterson for her assistance in the design and artistic rendering of Fig. 5.

**Financial support.** This work was supported by the United States Department of Agriculture National Institute of Food and Agriculture (award no. 2021-67019-34241) and the Colorado State University Graduate Degree Program in Ecology Small Grant Program.

**Review statement.** This paper was edited by Claudio Zaccone and reviewed by two anonymous referees.

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

## Remarks from the typesetter

TS1     Please note that it is not possible to only center "–". The first row is always left aligned.

TS2     At this stage of paper we cannot correct values. Are these values somewhere else in the paper to find?