# Peer review of "Shifts in controls and abundance of particulate and mineralassociated organic matter fractions among subfield yield stability zones."

_EGUsphere, 2023_

## Author Comment (AC1)

**REVIEWER 1**

These are my comments about the paper 'Shifts in controls and abundance of particulate and mineral associated organic matter fractions among subfield yield stability zones', from Sam J. Leuthold et al., submitted to EGUsphere.

It is a good paper, and actually I did not expect otherwise, owing to the already known experience and scientific proficiency of some of the senior authors. Therefore, as you will see, I have very little to add to the text. I have a few comments, however, that may be of help to authors, or perhaps may suggest some improvements in the paper.

**RESPONSE:** We appreciate the referee's kind comments about the paper, and their efforts in providing suggestions to improve the quality of the manuscript. We have read through and considered each of the comments in turn and have provided our response to them below.

**COMMENTS**

**COMMENT:** As I understand, this research is done simultaneously to other ones, namely that of Fowler et al., which is repeatedly cited (as 'in-review': perhaps the term 'submitted' would have been more appropriate?). The fact that this additional information (about the sites, plant production and so on) is not yet available is a pity. However this is a minor drawback in this work.

**RESPONSE:** We agree that it was challenging to not have the paper by Fowler et al. available at the time of review. However, we are pleased to announce that this paper was published during the intervening time and can now be found at https://doi.org/10.1038/s41598-024-51155-y. In a revised manuscript, we will replace the citations that are listed as "in-review" to the new citation (i.e., Fowler et al., 2024), and add the following citation to the reference list:

Fowler, A., Basso, B., Maureira, F., Millar, N., Ulbrich, R., and Brinton, W. F.: Spatial patterns of historical crop yields reveal soil health attributes in US Midwest fields, Sci Rep, 14, 465, https://doi.org/10.1038/s41598-024-51155-y, 2024.

**COMMENT:** Line 93. 'Zea mays', not 'Zea maize'.

**RESPONSE:** Thank you for pointing this out, we will correct this in the revised manuscript.

**COMMENT:** Line 145. Remove one of the two consecutive 'the'.

**REPSONSE:** Thank you for finding this typo, we will correct this in the revised manuscript.

**COMMENT:** Lines 252-255. If I well understood, your results seem contradictory with those of Maestrini & Basso. This is a very interesting result: could you stress it a bit more in the text? It goes almost unnoticed.

**RESPONSE:** This is a good point, and one that we believe needs to be further clarified in the revised manuscript to avoid any ambiguity. Our results are not in opposition to the work presented by Maestrini and Basso (2018). In the original text, we discuss the observation of both high-yielding, stable zones and unstable yield zones in low-lying areas in our study. However, we discussed the Maestrini and Basso work only in relation to the unstable yield zones in low-lying areas. This was ambiguous and requires clarification, as these findings are not contradictory-- Masestrini and Basso also find high-yielding, stable zones in low-lying areas in their observations, especially in rain-fed cropping systems. We are sorry to have omitted this finding from their work in our discussion, generating confusion, and we are happy to further clarify as necessary. We will address this comment by revising the text as follows—

**Original Text:** "As characterized by Maestrini and Basso (2018), unstable yield zones are often located in depositional areas that receive downslope contributions of fine soil particles on the decadal time scale (Ampontuah et al., 2006; Thaler et al., 2021). In our study, high-yielding, stable zones had soils with relatively high fine particle content (Fig. 3) and were found mostly in lower areas of the field as well (Fig. 4), which reflects previous observations of crop yield heterogeneity in areas of topographic complexity (Kravchenko and Bullock, 2000; Leuthold et al., 2021)".

**Revised Text:** "As observed by Maestrini and Basso (2018), both high-yielding, stable zones and unstable zones are often located in depositional areas that receive downslope contributions of fine soil particles on the decadal timescale. Our study finds a similar distribution of stability zones amongst low-lying areas, with both the high-yielding, stable zones and the unstable zones having a relatively high fine particle content (Fig. 3), and being found primarily in lower areas of the field (Figure 4). Our findings thus corroborate previous observations of crop yield heterogeneity in areas of topographic complexity (Kravchenko and Bullock, 2000; Maestrini and Basso, 2018; Leuthold et al., 2021)".

**COMMENT:** Lines 262-263. Extremely interesting finding, even though it has been observed before (as mentioned in line 265). Note, however, that the

relationship is very weak, in the very limit of signification (p = 0.048). Please mention this detail.

**RESPONSE:** The referee is correct that this relationship is somewhat muddled by the variability in cropping system, site characteristics, and climate, leading to the small $r^2$ value as well as the marginal significance. In the revised manuscript we will acknowledge these factors more explicitly so as not misrepresenting the strength of the relationship, revising the text as follows.

**Original Text**: In addition to these edaphic controls, we also observed an association between increasing yields and increasing MAOM-C content. Linear regression analysis indicated that as mean yield within a stability zone increased, so did MAOM-C (p = 0.048, r2 = 0.08; Supplemental Fig. 5). This finding mirrors recent studies that support causal linkages between increasing productivity and increased MAOM-C (Prairie et al., 2023; King et al., 2023; Hansen et al., in-revision).

**Revised Text:** In addition to these edaphic controls, we also observed an association between increasing yields and increasing MAOM-C content. Linear regression analysis indicated that as mean yield within a stability zone increased, so did MAOM-C. This relationship was weak, likely reflecting the influence of cropping system, soil physicochemical properties, and climate on variability in MAOM-C and yield (p = 0.048, r2 = 0.08; Supplemental Fig. 5). However, this result does mirror recent studies that support causal linkages between increasing productivity and increased MAOM-C (Prairie et al., 2023; King et al., 2023; Hansen et al., in-revision).

**COMMENT:** About the Figure 6. This figure summarizes, to some extent, the results of this experiment. It is nice. Note, however, that at a first glance, it is a bit unconsistent with some of the results mentioned before. For instance: POM-C is, apparently, the most relevant factor in determining Yield stability. Nevertheless, the previous text (lines 274-276, also the following ones) rather suggests that the relationship between POM-C and yield stability is unclear. The key is, perhaps, the sentence(s) 'POM-C content in unstable zones was significantly higher than in all stable zones (p = 0.019), which had the same POM-C content independent of yields (Fig. 2C)' (lines 273-274), which suggest rather a negative relationship: POM-C relates negatively to yield stability. In line 284 you state 'we did not observe evidence that POM-C conferred additional stability to cropping systems'. But perhaps the key is in your further sentences (lines 285 and following) that rather suggest that POM-C relates indirectly to yield stability. The reasons that make unstable the crop yield have, as a secondary result, the accumulation of POM (if I well understood). However the heading of figure 6 ('Relative importance of

variables in determining yield stability zones') rather suggests that you see POM-C as a cause of yield stability. May Figure 6 give an inexact view about your results? Is there any way to distinguish between 'likely causes' and 'likely consequences' of yield unstability?

**RESPONSE:** Yes, the referee is correct about this interpretation of the results relating POM-C concentrations to yield stability zone. While our initial hypothesis was that increasing POM-C would lead to increased yield stability, we found instead that unstable zones had significantly higher POM-C than any stable zone, regardless of yield. Our interpretation of these results is consistent with the referee's understanding—the conditions that lead to yield instability and annual fluctuations in yield are also conditions that would lead to the preservation of POM in these areas (i.e., anoxic soil conditions, increased residue inputs). The results presented in figure 6 are the relative importance of variables in predicting whether a zone would be stable or not via a Random Forest (RF) model. The values are computed based on the number of times a given variable is used to split nodes within an RF model, with higher values indicating more influence on the prediction outcome. The referee makes a good point that the figure caption and title could be misleading. In our revision we will reword the caption by replacing "determining," such that it does not refer to a causal mechanism (see below).

**COMMENT:** An additional detail: because all bars are given in a left-right orientation, it is not possible to distinguish between factors that affect (or are related) positively to yield stability, and those that affect (or are related) negatively to yield stability (which would be the case of POM-C, by the way). Would it be possible to distinguish them? For instance: perhaps the factors that affect negatively could be in the right-left orientation?

**RESPONSE:** Unfortunately, unlike linear models that have positive and negative coefficients that indicate the direction of a relationship, the variable importance metric does not have an inherent directionality. As detailed above, it instead reflects the structure of the RF model. We will change the figure caption to make this explicit upon revision as detailed below.

**Original Text:** Figure 6 – Relative importance of variables in determining yield stability zones as determined by the gradient boosted random forest model employed here. Relative importance represents the average of the feature importance over the course of 1000 model iterations. (POM: Particulate organic matter… GDD: Growing degree days).

**Revised Text**: Figure 6 – Relative importance of variables in predicting yield stability zone as determined by the gradient boosted random forest model

employed here. Relative importance represents the average of the feature importance over the course of 1000 model iterations and does not imply a directionality or causal linkage. (POM: Particulate organic matter… GDD: Growing degree days).

**FIGURES**

**COMMENT:** Figure 2. Just a question. I observed that the several points for a given 'Stability Zone' are not aligned horizontally. Is this deliberate, to facilitate a good view of these points (thus avoiding their superposition), or does this lack of alignment reflect some property of the points' sets?

**REPSONSE:** Yes, the "jittering" of the points was done to avoid excessive overall and improve readability of the figure. We will revise the figure caption as follows to make this explicit.

**Original Text:** Figure 2 – Normalized concentrations of total soil organic carbon (SOM-C) (a.), mineral associated organic matter carbon (MAOM-C) (b.), and particulate organic matter carbon (POM-C) (c.) among the various stability zones. Different colored points represent different farms. To account for edaphoclimatic differences among farms, we scaled all data using z-scores prior to analysis, with the mean and standard deviation calculated at the farm level, yielding a unitless metric to compare by. Different lower-case letters indicate significant differences ($p < 0.05$).

**Revised Text:** Figure 2 – Normalized concentrations of total soil organic carbon (SOM-C) (a.), mineral associated organic matter carbon (MAOM-C) (b.), and particulate organic matter carbon (POM-C) (c.) among the various stability zones. Different colored points represent different farms. To account for edaphoclimatic differences among farms, we scaled all data using z-scores prior to analysis, with the mean and standard deviation calculated at the farm level, yielding a unitless metric to compare by. Different lower-case letters indicate significant differences ($p < 0.05$). Points are offset horizontally to improve readability of the plot.

**COMMENT:** Figure 5. Should be improved. I noticed that, in the small icons, there is an area in light violet colour, and another in a blue-greenish colour. The relative area of each one changes. I deduce that the blue-greenish means 'unfair condition', but it is not clear. The legend of the figure does not say anything about it: the meaning of these two colours should be added to the legend, otherwise the precise meaning of the figure remains unclear. Besides this problem, these small icons may be impossible to read in a printed version: would it be possible to enlarge them a bit?

**RESPONSE:** This is a wonderful point, and one we did not consider during the initial iteration of this figure. We will update the figure in the revised manuscript to include a legend for the conditions within the icon boxes that makes the interpretation explicit. We will also increase the size of the icons and text throughout the figure to improve the readability of the figure, especially when printed.

**COMMENT:** Figure 6. Nice figure. That said, please correct 'Mehlic' to 'Mehlich'. See also my previous comments about this figure, which perhaps summarizes the whole results of this paper.

**RESPONSE:** We appreciate the reviewer pointing out this typo, and will fix it upon revision. In addition, please see our response above in further reference to this figure.

**REFERENCES**

**COMMENT:** The following cites are missing from the 'references' section:

Castellano et al 2015

Just et al 2023

King et al 2023

Prairie et al 2023

Van oost and Six 2023

**RESPONSE:** We apologize for the oversight and will ensure that the references section is complete upon revision. The citations for the papers the referee indicated will be added and are listed here:

Castellano, M. J., Mueller, K. E., Olk, D. C., Sawyer, J. E., and Six, J.: Integrating plant litter quality, soil organic matter stabilization, and the carbon saturation concept, Glob Change Biol, 21, 3200–3209, https://doi.org/10.1111/gcb.12982, 2015.

Just, C., Armbruster, M., Barkusky, D., Baumecker, M., Diepolder, M., Döring, T. F., Heigl, L., Honermeier, B., Jate, M., Merbach, I., Rusch, C., Schubert, D., Schulz, F., Schweitzer, K., Seidel, S., Sommer, M., Spiegel, H., Thumm, U., Urbatzka, P., Zimmer, J., Kögel-Knabner, I., and Wiesmeier, M.: Soil organic carbon sequestration in agricultural long-term field experiments as derived from particulate and mineral-associated organic

matter, Geoderma, 434, 116472, https://doi.org/10.1016/j.geoderma.2023.116472, 2023.

King, A. E., Amsili, J. P., Córdova, S. C., Culman, S., Fonte, S. J., Kotcon, J., Liebig, M., Masters, M. D., McVay, K., Olk, D. C., Schipanski, M., Schneider, S. K., Stewart, C. E., and Cotrufo, M. F.: A soil matrix capacity index to predict mineral-associated but not particulate organic carbon across a range of climate and soil pH, Biogeochemistry, https://doi.org/10.1007/s10533-023-01066-3, 2023.

Prairie, A. M., King, A. E., and Cotrufo, M. F.: Restoring particulate and mineral-associated organic carbon through regenerative agriculture, Proceedings of the National Academy of Sciences, 120, e2217481120, https://doi.org/10.1073/pnas.2217481120, 2023.

Van Oost, K. and Six, J.: Reconciling the paradox of soil organic carbon erosion by water, Biogeosciences, 20, 635–646, https://doi.org/10.5194/bg-20-635-2023, 2023.

---

## Author Comment (AC2)

The paper 'Shifts in controls and abundance of particulate and mineral associated organic matter fractions among subfield yield stability zones' deals with understanding the relationship between SOM and yield heterogeneity.

**COMMENT:** The repeated citation of a paper under review/submitted is not a minor drawback in this work. The accessibility of the information about the experimental design is lacking, making the paper really hard to understand. This "minor drawback" is even more exacerbated referring to other papers under review.

**RESPONSE:** We understand the inherent challenge of reviewing a paper that depends on materials that are being reviewed during the same period. However, we would like to point out that the key information regarding the experiment, including the methodology for the delineation of the yield zones, soil sampling approach, and soil analysis and processing methods were described in our manuscript, and relied on the work by Fowler et al. (2024) only for further detail we did not believe was necessary to the understanding of the work we present, but may be of interest to the reader. Further, of the three papers that are cited as, "In-Review", two have been published during the review period for this manuscript, including the Fowler et al. work. In the revised manuscript, we will update these citations, and we will remove any other "In-Review" citations from the text (i.e., Leuthold et al., in-review) unless is published in the meantime.

**COMMENT:** Moreover, the references section is incomplete and, among the citations, too many papers are authored or coauthored by the same authors of this paper.

**RESPONSE:** We apologize for any errors in the reference section and will update it with accurate citations upon revision. We have also provided the citations which were not present in the original text below:

Castellano, M. J., Mueller, K. E., Olk, D. C., Sawyer, J. E., and Six, J.: Integrating plant litter quality, soil organic matter stabilization, and the carbon saturation concept, Glob Change Biol, 21, 3200–3209, https://doi.org/10.1111/gcb.12982, 2015.

Just, C., Armbruster, M., Barkusky, D., Baumecker, M., Diepolder, M., Döring, T. F., Heigl, L., Honermeier, B., Jate, M., Merbach, I., Rusch, C., Schubert, D., Schulz, F., Schweitzer, K., Seidel, S., Sommer, M., Spiegel, H., Thumm, U., Urbatzka, P., Zimmer, J., Kögel-Knabner, I., and Wiesmeier, M.: Soil organic carbon sequestration in agricultural long-term field experiments as derived from particulate and mineral-associated organic

matter, Geoderma, 434, 116472,
https://doi.org/10.1016/j.geoderma.2023.116472, 2023.

King, A. E., Amsili, J. P., Córdova, S. C., Culman, S., Fonte, S. J., Kotcon, J.,
Liebig, M., Masters, M. D., McVay, K., Olk, D. C., Schipanski, M., Schneider,
S. K., Stewart, C. E., and Cotrufo, M. F.: A soil matrix capacity index to
predict mineral-associated but not particulate organic carbon across a
range of climate and soil pH, Biogeochemistry,
https://doi.org/10.1007/s10533-023-01066-3, 2023.

Prairie, A. M., King, A. E., and Cotrufo, M. F.: Restoring particulate and
mineral-associated organic carbon through regenerative agriculture,
Proceedings of the National Academy of Sciences, 120, e2217481120,
https://doi.org/10.1073/pnas.2217481120, 2023.

Van Oost, K. and Six, J.: Reconciling the paradox of soil organic carbon
erosion by water, Biogeosciences, 20, 635–646, https://doi.org/10.5194/bg-
20-635-2023, 2023.

We also appreciate that there is a need for diversity in the references
within the manuscript and can understand the reviewers concern that the
paper as it stands now is overly self-referential. Upon revision we will add
additional references that better dilute the citation pool and add further
credence to the points made in the paper.

**COMMENT:** Another main drawback relates to the analysis performed, whereas organic
C has been analyzed in the bulk soil, the fractions have been characterized
only for total C content. It would have been useful to have data on the
organic C content also in the fractions.

**RESPONSE:** The C values listed for the fractions do reflect organic C values, a fact
which we will make explicit upon revision. As none of the samples in our
sample set contained inorganic soil carbon (pH range from 5.68 – 6.49 and
confirmed with spot testing via application of HCl), the reported values for
total carbon in the fractions are interchangeable with organic carbon
values. We would revise the text in Lines 138 – 140 as below upon revision:

**Original Text:** After weighing, samples were ground to a fine powder using a mortar and
pestle and analyzed for C and nitrogen (N) concentrations via a VELP
CN802 Carbon Nitrogen Analyzer (VELP Scientific, Deer Park, NY).

**Revised Text:** After weighing, samples were ground to a fine powder using a mortar and
pestle and analyzed for C and nitrogen (N) concentrations via a VELP
CN802 Carbon Nitrogen Analyzer (VELP Scientific, Deer Park, NY). As

the soils contained no inorganic C, total C values obtained through elemental analysis reflect fraction organic C.

**COMMENT:** Moreover (Line 116) the authors should explain what the "Shimadzu method" is. The authors put a reference (Shimadzu, 2021) that is not available in the reference list. Are the authors sure this is a reference?

**RESPONSE:** We agree with the referee that the description of the organic carbon concentration in the bulk soil was incomplete and apologize for this oversight. In a revised manuscript, we will provide further details about the methodology and model, as detailed below—

**Original Text:** "Soils were analyzed for a range of properties including total soil organic C using the Shimadzu method with 900 °C combustion (Shimadzu, 2021), soil pH using a 1:1 soil:water extract and pH electrode method (Horton, 1995), Mehlich I and Mehlich III extracted nutrients (NCERA-13, 2015), and cation exchange capacity (Horton, 1995; NCERA-13 2015)."

**Revised Text:** "Soils were analyzed for a range of properties including soil pH using a 1:1 soil:water extract and pH electrode method (Horton, 1995), Mehlich I and Mehlich III extracted nutrients (NCERA-13, 2015), and cation exchange capacity (Horton, 1995; NCERA-13 2015). Total soil organic C was measured via dry combustion at 900 °C using a Shimadzu TOC-L coupled to a Solid Sample Dry Combustion Module SSM-5000A (Shimadzu Corporation, Kyoto, Japan), following manufacturer protocols (Shimdazu, 2017)."

Shimadzu. Total Organic Carbon Analysis. #638-94605C. Shimadzu User Manual. Shimadzu Scientific Instruments, Columbia MD. 2017.

**COMMENT:** Another analytical concern: how the authors measured the texture in a not direct way? And why?

**RESPONSE:** We chose to separate the texture into coarse and fine particles based on wet-sieving at 53 µm for a number of reasons. For one, we had a limited soil mass on which to complete our analyses (30-50 grams per plot). As texture measurement via the hydrometer method requires at least 40 grams of soil, we opted to only separate based on the size cut-off for sand grains, which coincided with our fractionation procedure. More conceptually though, as we assume that a first-order control on the concentration of mineral associated organic matter fraction in soil is the sum of the silt and clay particles (as demonstrated in Georgiou et al. (2022), Begill et al. (2023) and others), we were primarily interested in the abundance of this particle size fraction, rather than the explicit soil texture in our analysis. By

understanding the relative proportion of silt + clay particles, we could begin to understand the sorption potential of the different yield stability zones and its interactions with erosive potential and landscape topography to determine MAOM concentrations. In a revised manuscript we will make this point more explicit, changing the text in lines 118 – 120 to specify our interest in the proportion of fine particles, rather than individual texture classes.

**COMMENTS:** The figure 5, that should present the core results of the paper, contains a legend that is not informative at all. The authors used different colors, without really clarifying the meanings.

**REPSONSE:** We agree that the conceptual figure can be improved, especially in regard to providing further information about the colors within the icons and what they indicate. As also specified in the response to reviewer 1, upon revision we plan to make substantial changes to this figure, including creating a more thorough legend, increasing the text size, and generally improving readability.

**References**

Georgiou, K., Jackson, R. B., Vindušková, O., Abramoff, R. Z., Ahlström, A., Feng, W., Harden, J. W., Pellegrini, A. F. A., Polley, H. W., Soong, J. L., Riley, W. J., and Torn, M. S.: Global stocks and capacity of mineral-associated soil organic carbon, Nat Commun, 13, 3797, https://doi.org/10.1038/s41467-022-31540-9, 2022.

Begill, N., Don, A., and Poeplau, C.: No detectable upper limit of mineral-associated organic carbon in temperate agricultural soils, Global Change Biology, 29, 4662–4669, https://doi.org/10.1111/gcb.16804, 2023.